# Multi-Task Learning for Joint Indoor Localization and Blind Channel Estimation in OFDM Systems

**DOI:** 10.3390/s25134095

**Published:** 2025-06-30

**Authors:** Maria Camila Molina, Iness Ahriz, Lounis Zerioul, Michel Terré

**Affiliations:** Conservatoire National des Arts et Métiers, CEDRIC, 292 rue Saint Martin, 75141 Paris, France; iness.ahriz@lecnam.net (I.A.); lounis.zerioul@lecnam.net (L.Z.); michel.terre@lecnam.net (M.T.)

**Keywords:** indoor localization, fingerprint localization, channel state information, blind channel estimation, OFDM

## Abstract

In contemporary wireless communication systems, achieving precise localization of communicating devices and accurate channel estimation is crucial for enhancing operational efficiency and reliability. This study introduces a novel approach that integrates the localization task and channel estimation into a single framework. We present a multi-task neural network architecture capable of simultaneously estimating channels from multiple base stations in a blind manner while estimating user terminal coordinates in given indoor environments. This approach exploits the relationship between channel characteristics and spatial information, using the same channel state information (CSI) data to perform both tasks with a single model. We evaluate the proposed solution, assessing its effectiveness across differing antenna spacing configurations and indoor test environments using both WiFi and 5G orthogonal frequency-division multiplexing (OFDM) systems. The results show performance benefits, achieving comparable channel estimation results to other studies while simultaneously providing a localization estimate, resulting in reduced model overhead while leveraging spatial context. The presented system demonstrates potential to improve the efficiency of communication systems in real-world applications, aligning with the goals of emerging integrated sensing and communication (ISAC) systems. Results based on experimental data using the proposed solution show a 50th percentile localization error of 1.62 m for 3-tap channels and 0.89 m for 10-tap channels.

## 1. Introduction

The rapid advancement of wireless communication technologies and the ubiquity of the Internet of Things (IoT) in the modern technological landscape have significantly increased the availability of communication data in environments equipped with radio technologies, such as WiFi, Bluetooth, and 5G. This, consequently, has boosted the demand for location-based services (LBSs), which play a crucial role in various commercial and industrial applications, including asset tracking in smart factories, patient monitoring in hospitals, first-responder tracking, and indoor turn-by-turn navigation for autonomous systems. As indoor environments have become more connected, accurate, and efficient localization has become a fundamental component of modern communication systems [1].

Unlike outdoor positioning, which primarily relies on global navigation satellite systems (GNSSs), indoor localization is challenging due to multipath interference, signal attenuation, and the lack of a direct line of sight (LoS) with satellites [1]. To address these issues, researchers have explored alternative positioning techniques, leveraging wireless local area networks (WLANs), Bluetooth, radio-frequency identification (RFID), and low-power wide-area networks (LPWANs) such as LoRa. These solutions typically rely on geometric methods such as lateration and angulation [2], which estimate spatial parameters using time-of-flight (ToF), received signal strength (RSS), and angle-of-arrival (AoA) measurements [3]. While effective, these methods require precise signal propagation models or strict synchronization, making them impractical for dynamic indoor environments.

A widely used approach in indoor localization is fingerprinting, where machine learning and pattern-matching algorithms compare real-time communication parameter measurements against a pre-recorded dataset [4]. Among the different signal metrics used for fingerprinting, channel state information (CSI) has emerged as a powerful tool due to its fine-grained representation of channel characteristics, offering more robust and accurate positioning compared to traditional RSS-based methods [5]. CSI provides detailed phase and amplitude information, making it an attractive metric for WiFi, 5G, and future 6G-based localization solutions.

CSI is also essential for optimizing wireless networks. It enables beamforming, adaptive modulation and coding (AMC), and efficient resource allocation, all of which contribute to improved network performance [6]. More importantly, CSI can be exploited for precise device positioning, particularly in GNSS-denied environments, such as smart buildings, underground facilities, and dense urban areas [7].

WiFi- and Bluetooth-based localization systems have gained popularity due to their ubiquity in smart environments. However, the advent of multi-carrier communication technologies such as 5G and future 6G networks offers new opportunities for localization. These technologies provide higher bandwidths, massive antenna arrays (Massive MIMO), and dense access point deployments, enabling more precise positioning through advanced spatial processing techniques, such as AoA estimation, ToF analysis, and deep learning-based fingerprinting [8].

As localization capabilities become integrated into communication standards (e.g., 3GPP, IEEE) [8], researchers are investigating methods to optimize localization and communication simultaneously. However, one key challenge in CSI-based localization is channel estimation, which plays a crucial role in accurately interpreting CSI measurements. Traditional channel estimation relies on pilot symbols, which introduce overhead and reduce spectral efficiency, limiting the scalability of localization solutions [9].

Despite the potential of CSI for localization, current systems treat channel estimation and localization as separate tasks, leading to redundant computations and inefficiencies [10]. This separation increases computational complexity and limits real-time localization applications, especially in resource-constrained IoT environments.

To address these limitations, this work proposes a joint learning framework that integrates blind channel estimation and indoor localization into a single model. By leveraging a multi-task learning framework, we exploit the relationship between channel characteristics and spatial positioning to simultaneously estimate both the user location and the propagation channel without the need for dedicated pilot signals. As such, the proposed approach reduces computational overhead while improving localization accuracy, aligning with the vision of integrated sensing and communication (ISAC) in next-generation wireless networks [11].

Several studies have investigated machine learning solutions for either channel estimation or localization, but few have explored the inherent relationship between the two tasks [12]. Our proposed solution addresses this gap by using a multi-task neural network (MT-NN) capable of estimating the CSI-based position of a user while simultaneously performing blind channel estimation. This shared representation improves generalization, reduces redundancy, and provides a scalable solution for real-time CSI-based localization [13].

Non-deep learning-based approaches have also been proposed for multi-task solutions in OFDM systems. For instance, a tensor decomposition-based method has been introduced for extremely large-scale MIMO-OFDM systems employing dynamic metasurface antennas, which enables efficient channel parameter extraction with reduced training overhead [14]. Unlike our approach, this method relies on algebraic tensor factorization and does not utilize data-driven learning, making it complementary to our deep learning-based design.

The remainder of this paper is organized as follows: In Section 2, the state-of-the-art methods in indoor location and blind channel estimation using deep learning solutions are reviewed. Section 3 presents the formulation of the problem. In Section 4, the proposed solution and the system model are described in detail and the experimental setup and the different datasets used are presented. In Section 5 the performance of the solution and the obtained results are analyzed, and Section 7 concludes the paper.

## 2. Related Work

RSS-based indoor localization remains one of the most widely used techniques due to its simplicity and compatibility with existing wireless networks. However, the accuracy of RSS-based methods is often limited by interference, multipath fading, and fluctuations in signal strength, especially in dynamic indoor environments. To address these limitations, CSI-based localization methods have gained popularity, as they capture detailed information about phase and amplitude, allowing for more accurate localization.

A key challenge in CSI-based localization is channel estimation, which typically requires pilot signals or predefined reference data to estimate the wireless channel. Recent research has focused on blind channel estimation techniques, which eliminate the need for pilot signals and instead estimate channel parameters directly from the received signal. This approach significantly improves spectral efficiency and reduces computational complexity by eliminating the need for separate pilot transmission. Blind channel estimation is particularly advantageous in dynamic environments, where traditional methods may struggle to adapt to rapidly changing conditions.

One notable study introduces a multi-task learning framework that integrates blind channel estimation and localization into a single model. This approach eliminates the need for pilot-based CSI measurements, which are traditionally used for channel estimation. By leveraging blind estimation techniques, the model jointly optimizes both tasks, reducing computational overhead and improving localization accuracy. This method not only enhances spectral efficiency but also demonstrates robust performance in real-world environments where multipath and noise conditions are prevalent. By combining both tasks into a unified framework, the approach provides a scalable solution for large-scale IoT localization systems [15].

Another significant contribution in this area combines deep learning with blind channel estimation. The study uses convolutional neural networks (CNNs) to learn the complex patterns in the CSI data, which improves both channel estimation and localization simultaneously. By utilizing blind channel estimation, this method does not rely on pre-calibrated reference data, making it adaptable to changing environments. The use of deep learning techniques allows the system to automatically learn relevant features from the raw CSI data, improving localization accuracy even in multipath-rich environments. The proposed model achieves superior localization accuracy by addressing both the challenges of channel estimation and localization in non-line-of-sight (NLoS) conditions, which are common in indoor environments [16].

In addition, ref. [17] enhances the performance of blind channel estimation by integrating statistical methods such as visibility graph analysis. Visibility graph analysis helps to model the propagation environment by creating a network of signal paths. This method captures the spatial relationships between signal paths, providing a more robust and accurate estimation of both channel parameters and device position. When combined with blind channel estimation, visibility graph analysis improves the system’s ability to handle noisy environments and enhances localization accuracy. This integrated approach has shown resilience to environmental noise and has been demonstrated to outperform traditional CSI-based methods in terms of classification accuracy and robustness to dynamic changes in the environment.

Despite the advancements in blind channel estimation, a challenge that remains is the computational complexity of these techniques. While blind estimation methods eliminate the need for pilot signals, they still require sophisticated signal processing techniques to estimate the channel accurately. These methods, although more efficient in terms of spectral usage, may incur higher computational costs due to the complexity of the algorithms involved. Recent efforts aim to reduce these costs by jointly optimizing localization and channel estimation, as a study that combined deep learning with blind channel estimation [16]. This joint optimization approach reduces the computational overhead, making real-time localization feasible even in large-scale IoT networks.

The approach proposed in this paper builds on these advancements by introducing a novel multi-task learning framework that integrates blind channel estimation and localization into a unified model. Unlike traditional methods, our approach leverages blind estimation to improve spectral efficiency and reduce computational overhead. By jointly optimizing both tasks, we achieve superior localization accuracy in both simulated (NYUSIM 5G) and real-world (WiFi CSI) environments. This integration of blind channel estimation and localization makes the system more scalable and adaptable to dynamic environments, aligning with the goals of integrated sensing and communication for next-generation IoT localization systems [15].

Unlike previous studies that rely on pilot-based CSI measurements or static RSS fingerprints, this approach offers several key advantages:It uses the raw received signals as input to the localization pipeline.It eliminates the need for pilot signals through blind estimation, thereby improving spectral efficiency and reducing the overhead associated with traditional pilot-based methods.The multi-task learning framework reduces computational overhead by jointly optimizing channel estimation and localization within the same model, making the system more efficient in terms of both computation and energy consumption.The proposed approach achieves superior localization accuracy in both simulated 5G environments (such as NYUSIM) and real-world WiFi CSI environments, demonstrating its effectiveness across different scenarios.

## 3. Problem Formulation

### 3.1. Indoor Localization

In indoor localization systems using wireless communication, different communication parameters can be used to establish a fingerprinting localization solution. While RSS-based solutions are widely adopted, the lack of stability that RSS measurements present highlights the need for more robust and fine-grained information. As such, indoor localization using CSI has received attention in recent years. The physical layer information that is observed captures the rich details of the wireless channel, such as multipath propagation effects, providing the amplitude and phase of the subcarriers [18,19].

To establish a fingerprint indoor localization solution using CSI, there are two implementation steps to be carried out. The first step is an offline phase, during which CSI data measurements are collected to build a reference database. For this, the indoor environment is either divided into a grid, with each grid cell treated as a distinct location, or *M* reference points (RPs) are defined at regular intervals. Furthermore, *N* access points (APs) or base stations (BSs) are placed in the environment at fixed positions to ensure sufficient signal coverage. The fingerprint database is built by collecting multiple measurements at a given RP from all of the available APs. We define a single measurement from $AP_{n}$ at location *m* as(1)$$H_{n}^{m}=\left[H_{1},H_{2},\ldots ,H_{K}\right]$$
with $H_{k}\in C$ representing the channel frequency response (CFR) for subcarrier *k*.

As such, when establishing the fingerprint base, the obtained fingerprint matrix is defined in Equation (Equation 2) and its corresponding coordinate matrix is defined in Equation (Equation 3), where multiple fingerprint measurements are collected at each RP. Let the matrix of all measurements at all locations and all access points be denoted as(2)$$H=\left[\begin{array}{cccc} H_{1}^{1} & H_{2}^{1} & \ldots & H_{N}^{1}\\ H_{1}^{2} & H_{2}^{2} & \ldots & H_{N}^{2}\\ \vdots & \vdots & \ddots & \vdots \\ H_{1}^{M} & H_{2}^{M} & \ldots & H_{N}^{M} \end{array}\right],$$
where

$H\in C^{M\times N\times K}$;*M* is the total number of RPs;*N* is the total number of access points;*K* is the number of subcarriers in each CSI measurement.

(3)$$P=\left[\begin{array}{c} p_{1}\\ p_{2}\\ \vdots \\ p_{M} \end{array}\right],$$
where

*M* is the total number of RPs;Each $p_{m}=\left[x_{m},y_{m}\right]\in R^{2}$ represents the 2D coordinates (with $x_{m}$ and $y_{m}$ being the *x*- and *y*-coordinates of the *m*-th fingerprint location).

Secondly, an online phase is defined, where the user terminal (UT) collects new CSI measurements from the APs in the indoor environment at a given, unknown position $\left(x_{\theta },y_{\theta }\right)$. To estimate these coordinates, the recorded fingerprint $\left(H_{1}^{\theta }\ldots H_{n}^{\theta }\right)$ is used for localization by applying a pattern-matching algorithm to compare it with the reference database.

### 3.2. Channel Estimation

An essential precursory step in locating a user terminal in an OFDM system is the accurate estimation of the propagation channels at the receiver.

#### 3.2.1. Data-Assisted Channel Estimation

Traditionally, channel estimation is carried out in a data-assisted manner. This method relies on pilot symbols, known values that are transmitted using several pilot subcarriers in order for the receiver to estimate the channel. We define the received OFDM symbol received from one BS at subcarrier *k* as(4)$$r_{k}=H_{k}s_{k}+w_{k}$$
with

$r_{k}$ the received symbol at subcarrier *k*;$s_{k}$ the transmitted symbol at subcarrier *k*;$H_{k}$ the *k*-th element of the channel frequency response;$w_{k}$ additive white Gaussian noise.

At the receiver, the pilot values and position indexes are known.

Let $s_{p}\in C^{P}$ be the vector of known transmitted pilot symbols at pilot subcarriers;$r_{p}\in C^{P}$ be the corresponding received symbols;$h_{p}\in C^{P}$ be the unknown channel frequency response values at the pilot subcarriers;
where *P* is the number of pilot subcarriers. It is possible to estimate the channel at the pilot subcarriers using a least squares estimate as defined in Equation (Equation 5).(5)$$\hat{h_{p}}=\arg\min_{h_{p}}{∥r_{p}-h_{p}s_{p}∥}_{2}^{2}$$

Subsequently, the estimated channel $\hat{h_{p}}$ at the pilot indexes can be used to interpolate the channel at the data subcarriers [5].

#### 3.2.2. Blind Channel Estimation

While the aforementioned data-assisted channel estimation solution provides a low-complexity approach, its efficiency decreases in high-noise scenarios. Alternative pilot-based channel estimation solutions exist, such as minimum mean-square error estimation (MMSE), which are more robust but increase the complexity of the estimation.

Blind channel estimation aims to estimate the channel without the use of pilot symbols, thus increasing the bandwidth available for data.

Solutions using statistical characteristics of the channel, higher-order statistics, and other properties of the signal have garnered much interest [20]. Nevertheless, in this work we use deep learning models to blindly estimate the channel and thus compare the performance in terms of the symbol error rate (SER).

As such, given the received signal of Equation (Equation 4) on all subcarriers from all APs, a neural network is trained to take as input the real and imaginary parts of each symbol in order to predict the unknown channel.

In this study, we consider a system with two base stations transmitting to a single user terminal equipped with one antenna in an OFDM setup (Figure 1). Our objective is to predict the position of the user terminal using the channel information from both APs as the fingerprint features without relying on pilot signals. Moreover, we explore the use of a joint model to perform blind simultaneous estimation of both channels. For indoor localization purposes, it is crucial to access certain communication parameters relevant to localization; we cannot solely rely on the received signal. Thus, we utilize an estimation of the channel frequency response for this purpose.

## 4. Materials and Methods

### 4.1. Proposed Solution

The proposed methodology builds upon the approach presented in [20] for the simultaneous estimation of two channels from different base stations in a scenario where no pilot symbols are transmitted.

We denote the CFRs corresponding to each AP as *H* and *G*, respectively. The inverse fast Fourier transform (IFFT) is applied to obtain the corresponding channel impulse response (CIR), *h*, and *g*. The values of the CIRs are truncated to extract the most significant elements and limit the amount of noise, and the obtained elements are normalized. The channel responses are then applied to the respective transmitted symbols, and noise is added at various signal-to-noise ratio (SNR) levels. The pair of transmitted signals is then combined to obtain the received signal, whose expression is defined in Equation (Equation 6).(6)$$r_{n_{s},k}=H_{k}s_{1,n_{s},k}+G_{k}s_{2,n_{s},k}+w_{n_{s},k}$$
where

$r_{n_{s},k}$ represents the $n_{s}$-th received symbol at subcarrier *k*;$s_{1,n_{s},k}$ is the transmitted symbol from the first transmitter for the $n_{s}$-th symbol at subcarrier *k*;$s_{2,n_{s},k}$ is the transmitted symbol from the second transmitter for the $n_{s}$-th symbol at subcarrier *k*;$H_{k}$ is the CFR for the first transmitter at subcarrier *k*;$G_{k}$ is the CFR for the second transmitter at subcarrier *k*;$w_{n_{s},k}$ is the noise associated with the $n_{s}$-th symbol at subcarrier *k*.

The mixed signal, corresponding to the observed signal at the receiver, is then used as the input for the algorithm detailed in Algorithm 1, which exploits the use of higher-order moments to carry out the distance-based sorting algorithm described in Algorithm 2.

In order to obtain the channel information from the received signal relevant to the localization task, the first stage of the receiver consists of carrying out the cyclic prefix (CP) cancellation followed by a fast Fourier transform (FFT). Taking into account the statistical properties of the used modulations and additive white Gaussian noise, we calculate $\sum_{n_{s}=0}^{N_{s}-1}{\left(r_{n_{s},k}\right)}^{4}$ and $\sum_{n_{s}=0}^{N_{s}-1}{\left(r_{n_{s},k}\right)}^{8}$. Combining the two equations, we obtain a second-degree equation whose unknown variable is either ${\hat{H}}_{k}^{4}$ or ${\hat{G}}_{k}^{4}$. Solving this leads to a set of 2K roots, and the primary challenge is determining which one is ${\hat{H}}_{k}^{4}$ and which one is ${\hat{G}}_{k}^{4}$. This presents the main limitation of the solution, as the initial study assumes at most two crossover points, resulting in $2^{2}$ possible combinations, as illustrated in Figure 2. Evaluating all possible combinations of crossovers between the frequency responses would quickly become intractable.
**Algorithm 1:** Preprocessing mixed channels.**Require:** 
Received OFDM signal r(t)**Ensure:** Initial estimation of CFRs H(f) and G(f)1: **for** each OFDM symbol r(t) **do**2:    $r_{\mathrm{CP}}\left(t\right)\leftarrow$ CP cancellation on r(t)3:    $R\left(f\right)\leftarrow \mathrm{FFT}(r_{\mathrm{CP}}\left(t\right))$4:    $a_{k}\leftarrow \frac{1}{N_{s}}\sum_{n_{s}=0}^{N_{s}-1}{\left(r_{n_{s},k}\right)}^{4}$5:    $b_{k}\leftarrow \frac{1}{N_{s}}\sum_{n_{s}=0}^{N_{s}-1}{\left(r_{n_{s},k}\right)}^{8}$6:    $A_{1}\leftarrow p_{18}-70\left(p_{14}^{2}\right)+\frac{\left(p_{14}^{2}\right)p_{28}}{\left(p_{24}^{2}\right)}$7:    $A_{2}\leftarrow 70a_{k}p_{14}-\frac{2a_{k}p_{14}p_{28}}{\left(p_{24}^{2}\right)}$8:    $A_{3}\leftarrow \frac{a_{k}^{2}p_{28}}{\left(p_{24}^{2}\right)}-b_{k}$9:    $\left(R_{1},R_{2}\right)\leftarrow \mathrm{roots}\left(A_{1},A_{2},A_{3}\right)$10:    $\left(H_{\mathrm{est}}^{4},G_{\mathrm{est}}^{4}\right)\leftarrow \mathrm{sort}\left(R_{1},R_{2}\right)$11: **end for**

**Algorithm 2:** Sorting algorithm used in [20].
**Require:** Estimated roots $\left(R_{1,k},R_{2,k}\right)$ for k=0,…,K−1**Ensure:** Sorted estimates $\left({\hat{H}}_{k},{\hat{G}}_{k}\right)$1: **Initialization: **${\hat{H}}_{0}\leftarrow R_{1,0}$, ${\hat{G}}_{0}\leftarrow R_{2,0}$2: **for** k=1 to K−1 **do**3:    **if** $\left|R_{1,k}-{\hat{H}}_{k-1}\right|+\left|R_{2,k}-{\hat{G}}_{k-1}\right|<\left|R_{2,k}-{\hat{H}}_{k-1}\right|+\left|R_{1,k}-{\hat{G}}_{k-1}\right|$ **then**4:        ${\hat{H}}_{k}\leftarrow R_{1,k}$, ${\hat{G}}_{k}\leftarrow R_{2,k}$5:    **else**6:        ${\hat{H}}_{k}\leftarrow R_{2,k}$, ${\hat{G}}_{k}\leftarrow R_{1,k}$7:    **end if**8: 
**end for**



The blind estimation method used here relies on the statistical properties of modulated signals and additive white Gaussian noise. Under these assumptions, the fourth- and eighth-order moments of the received signal at each subcarrier can be expressed in terms of the 4th and 8th powers of the individual channels $H_{k}$ and $G_{k}$. $a_{k}=\sum_{ns=0}^{N_{s}-1}{\left(r_{ns,k}\right)}^{4}$ estimates the aggregated 4th-order moment, which approximates $H_{k}^{4}p_{1,4}+G_{k}^{4}p_{2,4}$, where $p_{i,j}$ are modulation-dependent constants. $b_{k}=\sum_{ns=0}^{N_{s}-1}{\left(r_{ns,k}\right)}^{8}$ estimates the 8th-order moment, which includes mixed terms such as $H_{k}^{8},G_{k}^{8}$, and $H_{k}^{4}G_{k}^{4}$. These two moments are then used in lines to construct the coefficients of a quadratic equation whose roots represent the estimated values of $H_{k}^{4}$ and $G_{k}^{4}$. The exact derivation follows the approach detailed in [20], facilitating the transformation of the blind channel estimation problem into a tractable algebraic form.

To address this blind source separation problem, a machine learning-based solution is proposed, with a dual objective:Predict the position of the user terminal using uncertain channels as input.Denoise and untangle the channel frequency response pairs to estimate the channel.

This approach aims to precisely separate $H^{4}$ and $G^{4}$, ensuring clearer and more distinct identification of these components. By leveraging machine learning capabilities, the process will achieve higher accuracy in distinguishing between $H^{4}$ and $G^{4}$, leading to improved performance and reliability in handling the CFRs.

The proposed model for the task is a sequence-to-sequence model using stacked bidirectional long short-term memory (Bi-LSTM) layers [21]. Traditionally used for time-series data, these architectures are capable of modeling the complexities inherent to sequential data. The core mechanisms of long short-term memory (LSTM) cells are the cell state and gates, with the cell state acting as a path that allows relevant information to persist through time without passing through network layers. The LSTM gates are then responsible for altering this state: the forget gate decides which information to discard, updating the cell state as $C_{t-1}^{\prime }=f_{t}\times C_{t-1}$, where $f_{t}$ is an attenuation factor calculated as described in Equation (Equation 7); the input gate determines new information to be added, generating candidate values as shown in Equation (Equation 9) and updating the state ($C_{t}=C_{t-1}^{\prime }+u_{t}\times {\tilde{C}}_{t}$), where $u_{t}$ is defined by Equation (Equation 8). The output gate computes the new hidden state.(7)$$f_{t}=\sigma \left(W_{f}\cdot \left[h_{t-1},x_{t}\right]+b_{f}\right)$$(8)$$u_{t}=\sigma \left(W_{u}\cdot \left[h_{t-1},x_{t}\right]+b_{u}\right)$$(9)$${\tilde{C}}_{t}=\tanh\left(W_{c}\cdot \left[h_{t-1},x_{t}\right]+b_{c}\right)$$(10)$$h_{t}=\tanh\left(C_{t}\right)\times \sigma \left(W_{o}\cdot \left[h_{t-1},x_{t}\right]+b_{o}\right)$$

This structure enables LSTM networks to handle sequential data by successively updating the states and outputs of the network based on current inputs and past hidden states.

Building on the LSTM cell, Bi-LSTM networks process sequences in both the forward and backward directions. The encoder module is made up of five stacked Bi-LSTM layers.

As the CFR values $H^{4}$ and $G^{4}$ are two complex-valued vectors, the real and imaginary elements of $H^{4}$ and $G^{4}$ were separated into different vectors. As such, the 4th-order channel estimation module is fed as input a 4 by *K* matrix, and its outputs are of equal size. Two proposed strategies are evaluated:A single-task channel estimation model: an encoder–decoder model made up of a bidirectional LSTM encoder and an LSTM decoder. The encoder processes the multivariate input sequence using four layers of Bi-LSTMs with dropout for regularization. The decoder, also made up of four LSTM layers with dropout, processes the latent representation to generate the sorted and denoised estimations of $H^{4}$ and $G^{4}$.A single-task localization model: an encoder–decoder model made up of a Bi-LSTM encoder followed by three fully connected layers predicting the receiver’s (x,y) coordinates.A multi-task model that builds on the single-task models by incorporating a fully connected localization head to the encoder–decoder architecture. For the localization task, three fully connected layers take as input the final hidden state of the decoder in order to predict the corresponding *x* and *y* coordinates of the UT. The objective of the multi-task learning approach is for the model to learn a shared representation of the data that is more expressive due to the jointly learned tasks and consequently improve the generalization capabilities of the model as well as the efficiency and performance in comparison to multiple specific models trained to solve a unique task [6].

The proposed multi-task model, depicted in Figure 3, follows a sequence-to-sequence structure. It is composed of the following components:Encoder: A 4-layer bidirectional LSTM, processing the input CSI sequence. Each layer applies dropout at a rate of 0.2 for regularization.Decoder: A 4-layer unidirectional LSTM, which refines the encoded features for CSI denoising and detangling.Channel estimation head: A linear layer maps the decoder output at each subcarrier index to four denoised and detangled CFR output vectors, for H and G, with their respective real and imaginary parts.Localization head: The final hidden state from the decoder is passed through three fully connected layers. The final layer predicts the receiver’s position by outputting the 2D coordinates (x, y).

The model is trained with a joint loss function:(11)$$L_{\mathrm{total}}=\alpha \cdot L_{\mathrm{channel}}+\beta \cdot L_{\mathrm{localization}}$$
where $L_{\mathrm{channel}}$ is the reconstruction loss of the denoised and detangled CFR vectors and $L_{\mathrm{localization}}$ is the localization loss, with both loss terms using the mean squared error (MSE) loss. Empirically, α=0.9 and β=0.1 yielded balanced convergence across tasks. Optimization is performed using the Adam optimizer, with a learning rate of 0.001, and a batch size of 64. All models were trained for a maximum of 500 epochs. Bi-LSTM layers are particularly well suited to CSI-based localization because CFR across frequency subcarriers exhibits structured, correlated behavior due to multipath propagation and channel coherence that is sequential in nature.

### 4.2. Experimental Setup

In this section, we present the different datasets and the environments used during the evaluation of the proposed solution. Within the framework of this research, we focused on two OFDM localization datasets.

#### 4.2.1. NYUSIM Dataset

To evaluate the proposed blind channel estimation and localization solution, the first dataset utilized is derived from the NYUSIM dataset, generated using the publicly available NYUSIM simulation tool [7]. This tool facilitates the creation of synthetic communication data by incorporating realistic indoor propagation parameters, LoS conditions, and shadow-fading effects, thereby closely replicating real-world environments. The signal data produced by this tool is both spatially and temporally consistent, ensuring high fidelity to actual conditions. Version 4.0 of the NYUSIM tool was specifically used for this purpose, facilitating the generation of an initial ad hoc dataset for indoor localization studies. The realistic simulation capabilities of NYUSIM make it an important resource for testing and validating new algorithms in controlled yet realistic settings, providing a robust foundation for the evaluation of the proposed solution.

The simulated environment was configured according to the parameters specified by 3GPP for an indoor hotspot in office settings, incorporating NLoS conditions and shadow-fading variations, simulating multipath and obstruction effects. The scenario includes two distinct APs, each with a single transmitting antenna. The APs are spaced 100 m apart. The UT, or receiver, is also equipped with one antenna and is assumed to move along a hexagonal track within the environment at a speed of 3.2 m per second. To generate the dataset, the simulation covered a total track distance of 36 m. Measurements from both base stations were recorded at 2 m intervals, resulting in a total of 18 measurement positions. At each position, 1000 measurements were simulated, providing a comprehensive dataset for analysis. This setup ensures a detailed and realistic evaluation of the proposed blind channel estimation and localization solution in a controlled yet representative environment.

#### 4.2.2. WiFi CSI Dataset

A secondary dataset consists of CFR measurements recorded during a collection conducted in a university laboratory in Paris. Figure 4 depicts the floor plan of the 15 m × 15 m laboratory, including a main corridor and several adjoining offices and meeting rooms. An HP laptop served as the signal transmitter, stationed on a table within the central office room. Operating in injection mode, the laptop transmitted intermittently at a rate of 100 packets per second. This setup proved highly effective for our laboratory environment; however, provisions are in place to incorporate multiple transmitters for future expansions.

In Figure 4, the blue dots represent the 70 training reference points spaced at one-meter intervals, while the 28 testing locations are marked with red squares. During the offline training phase, CSI measurements were collected by a Humming Board (HMB) Pro device at these reference points to construct a raw radio map. The receiver was static during data collection at each of the discrete locations. The dataset features moving personnel and dynamic conditions such as the opening and closing of doors as the environment was in regular use during the collection period. The transmitter and receiver were each equipped with 3 antennas, allowing for a MIMO collection scenario. Approximately 5000 CFRs were recorded at each reference point, stored as radio-frequency signatures within the device’s firmware. For the online phase, the HMB receiver was moved among the 28 testing locations to capture CSI packets of similar size. The receiver was positioned at the same height, establishing a straightforward 2D platform for precise indoor position estimation. This methodology ensures comprehensive data collection and accurate localization capabilities within our laboratory environment. The WiFi CFR measurements were collected in 2019 as part of the experiment described in [4].

To ensure a rigorous comparison and align with our simulated data, we adopt a setup with a single receiving antenna. The presence of two APs is simulated using two transmitting antennas. By using this setup, we aim to maintain consistency and effectively evaluate the performance of the proposed solution under conditions that mirror those encountered in the simulations. This approach ensures that the evaluations are robust and that the results obtained are directly comparable across different test scenarios. Furthermore, given the short distance between the transmitting antennas (approximately 15 cm), this setup allows for a detailed study of the impact of spatial diversity on the performance of the proposed solution. By analyzing how the proximity of the antennas influences the effectiveness of the neural network, we can gain deeper insights into its role in denoising and untangling the channel frequency response pairs. This helps to better understand the benefits and limitations of spatial diversity in enhancing the accuracy and reliability of the separation process.

## 5. Results

This section presents the results obtained from the integrated framework for joint localization and channel estimation. The framework’s processing pipeline is depicted in Figure 5. The results are presented in two sets for each dataset: the first set utilizes test measurements gathered at coordinates identical to those used for training and validation. The second set evaluates the model’s generalization capabilities by using data collected at positions different from those in the training dataset. These results aim to provide a comprehensive evaluation of the framework’s performance under both familiar and novel conditions. They assess its robustness and effectiveness in real-world scenarios, beyond the training positions.

### 5.1. Localization Results

To evaluate the performance of our proposed multi-task network for localization, we compare its results with different CSI-based fingerprint algorithms. We consider a weighted K-nearest neighbors (WKNN) algorithm, a traditional fully connected deep neural network (DNN), and the state-of-the-art localization solution iPos-5G [22], as these represent a range of traditional and advanced approaches in the field. The iPos-5G paper presents a deep learning indoor localization solution using 5G CSI. In the offline phase, a preprocessing step is applied to the fingerprints in order to reduce the CSI noise, applying cross-correlation analysis, wavelet denoising, and Hampel filtering. The denoised CFR data is used to train a denoising autoencoder, which learns a compressed representation of the fingerprint data. During the online phase, the test CSI samples are processed using the denoising steps and are then fed through the autoencoder model. The receiver’s position is estimated using a probabilistic model, and the denoised CFRs are compared to stored fingerprints using a supervised radial basis function (S-RBF) kernel to obtain a similarity metric. According to the study, iPos-5G improves accuracy by 16–37% when compared to alternative methods.

We also compare the use of the presented single-task LSTM localization model. To this end, all localization models were presented the initial mixed channel estimate obtained from Algorithm 1 as input features, testing their efficacy on the localization task.

To assess their performance, we evaluate the cumulative distribution functions (CDFs) of localization error presented in Figure 6 and Figure 7 for the NYUSIM and WiFi scenarios respectively.

The multi-task model demonstrates superior performance compared to the other evaluated strategies. This comes from its ability to integrate the localization task and channel denoising within one framework, thereby exploiting the use of spatial context. This integrated approach leads to significant improvements in the accuracy and efficiency of both localization and channel estimation, as discussed later in Section 5.2.

By embedding the localization task alongside channel estimation, the multi-task model makes use of shared information more effectively. This allows the model to capitalize on spatial relationships between transmitter locations and received signals, refining both estimation accuracy and localization precision. This approach highlights the benefits of incorporating diverse tasks within a unified machine learning architecture, ensuring robust performance in complex communication environments. Furthermore, the use of LSTM cells incorporates sequential context into the encoded representation of the data, modeling subcarrier dependencies, which are not considered in traditional neural networks.

To go deeper into performance analysis, Table 1 and Table 2 provide detailed information on the localization errors for both datasets. These preliminary results present the performance of the proposed method across various environmental conditions, considering two different levels of antenna spacing configurations and two distinct scenarios. Specifically, the method excels in both familiar and novel environments, where test measurements are either collected at the same coordinates as those used for training or at different locations. The results from the NYUSIM and WiFi datasets demonstrate the superiority of the multi-task model over single-task learning for indoor localization. In both datasets, the multi-task model consistently outperforms the single-task and iPos models, achieving lower mean, median, and 90th percentile errors, indicating improved robustness across different environments. For the NYUSIM dataset, the proposed solution reduces the mean localization error to 2.40 m, outperforming the single-task model (2.76 m) and achieving better accuracy than iPos (3 m), KNN (4.58 m), and DNN (3.69 m). Similarly, in the WiFi dataset, the presented multi-task model achieves a mean error of 2.49 m, improving on the single-task version (2.70 m) and surpassing iPos (3.16), KNN (4.09 m), and DNN (3.69 m). While KNN and iPos achieve the lowest minimum errors, their higher mean and percentile errors indicate inconsistency across test cases. The multi-task model’s lower 90th percentile errors (5.19 m in NYUSIM and 4.61 m in WiFi) further highlight its reliability in handling worst-case scenarios. These results confirm that multi-task learning improves feature generalization, leading to more accurate and stable indoor positioning compared to traditional machine learning and deep learning methods.

By evaluating these scenarios, the study highlights the method’s ability to adapt and generalize effectively. It not only achieves improved performance in scenarios where training and testing positions are the same but also demonstrates better generalization in scenarios where the evaluation positions differ from those seen during training. This versatility underscores the method’s reliability and applicability across different spatial contexts, confirming its potential for practical deployment in diverse real-world scenarios.

The iPos-5G system reports mean absolute errors (MAEs) of 2.32 m and 2.94 m in two distinct environments: an office area measuring 7.5 m by 16.5 m, and a corridor setting while using a single base station. These results demonstrate the effectiveness of iPos-5G in practical deployment contexts. However, the CFRs used by iPos-5G are obtained through several preprocessing steps and pilot-based channel estimation. In contrast, our proposed multi-task model processes raw, mixed signals in a blind environment without prior channel separation. Despite this added challenge, our model achieves comparable or improved accuracy compared to both the NYUSIM and WiFi datasets, demonstrating its robustness and the effectiveness of the joint learning approach in handling realistic, noisy conditions.

To further evaluate the model’s behavior under different noise conditions, Figure 8 shows how the median, 90th, and 95th percentile localization errors vary with SNR for the multi-task model on the NYUSIM dataset. The observed stability in both median and higher-percentile errors suggests that the model’s performance is not significantly affected by the SNR variation used during training and evaluation due to the incorporation of different levels of noise during the training phase of the model, as the model was trained on a dataset spanning multiple SNR levels (from 10 dB to 40 dB).

### 5.2. Blind Channel Estimation Results

To evaluate the performance in the blind channel estimation task for two channels simultaneously, we propose a comparison of the results obtained using the following channel estimation strategies:The baseline presented in [20], denoted as the initial solution.A single-task model using the same encoder–decoder architecture depicted in Figure 3, without the fully connected layers, specifically trained to perform the task of disentangling and denoising $H^{4}$ and $G^{4}$.The proposed multi-task solution trained to estimate $H^{4}$ and $G^{4}$ as well as predict the location of the receiver.

The presented results were obtained using pairs of channels drawn from the NYUSIM dataset, which consists of K=128 subcarriers. The proposed solutions were evaluated on a test subset of the collected data, ensuring that this subset was not seen during the training of the models to provide an unbiased assessment of performance. Specifically, 100 channel pairs were utilized to test the effectiveness of the channel estimation task. The evaluation used 250 OFDM symbols with 4-QAM modulation to simulate realistic conditions.

The results, as depicted in Figure 9, present the SER averaged over both signals. The inclusion of multiple channel pairs ensures that the results are robust and indicative of real-world performance, demonstrating the effectiveness of the proposed deep learning approach in denoising and untangling the CFR pairs.

The obtained results demonstrate that the multi-task model achieves comparable performance with a slight improvement over the initial blind estimation baseline. While the gains in SER are modest, they are noteworthy given that the same model also performs localization without the need for separate training or inference pipelines. This suggests that the shared encoder can capture useful features that benefit both tasks. By leveraging this additional spatial context, the multi-task model improves the accuracy and robustness of the estimations, outperforming single-task approaches. This improvement highlights the value of using location data to better understand and predict channel behavior, leading to more reliable communication system performance. The enhanced results validate the effectiveness of the multi-task learning framework in integrating diverse information sources to achieve more precise and efficient channel estimation.

## 6. Discussion

We restrict our evaluation to 4-QAM modulation since this work marks preliminary research on joint localization and blind channel estimation. This lets us concentrate on the fundamental behavior and efficiency of the suggested architecture free from the additional complexity of higher-order modulations. In future work, it would be valuable to investigate the model’s performance under different modulation schemes such as 16-QAM or 64-QAM to assess its scalability and robustness in more demanding scenarios.

While the results on both datasets show the potential for joint localization and estimation using a single model, there is room for improvement as current norms expect centimeter-level accuracy in pilot-based scenarios, where channels are estimated separately. To evaluate how localization accuracy can improve in blind scenarios, we studied the impact of channel length. For this, we trained the multi-task model on the NYUSIM dataset using channels with increasing lengths: 3-tap, 5-tap, 10-tap, and 30-tap. The results of these are presented in Table 3. The results show that increasing the channel length improves localization performance, as longer channels capture more expressive information about the propagation environment.

The results show that increasing the channel length improves localization performance. Across all statistical metrics, the mean localization error decreases significantly, from 2.40 m for the 3-tap channel to 0.79 m for the 30-tap channel. Similarly, the median error drops from 1.62 m to 0.58 m. In edge cases, the maximum error and 90th percentile error reduce significantly, as illustrated by the CDF in Figure 10.

The obtained results show the positive impact of channel length on improving localization performance. Longer channels contain more expressive information, capturing multipath and propagation characteristics that allow for a richer representation of the environment to be learned by the neural network’s encoder. These results suggest that, in scenarios where it is possible, increasing the channel length can be an effective strategy for improving the accuracy of indoor localization solutions. However, the trade-off between localization accuracy and channel estimation must also be considered for practical deployment.

### Performance Analysis

To assess the computational efficiency of our proposed solution, carrying out both localization and channel estimation, we measured the average inference time per sample over 100 samples on a Mac M1 system. We compare the results against the baseline channel estimation solution with a single-task localization model, a single-task channel estimation model with a single-task localization model, and finally, the multi-task channel estimation and localization model. The results are as follows.

As shown in Table 4, the proposed multi-task model achieves the lowest inference time, despite simultaneously performing both localization and channel estimation. Unlike the baseline approach, which requires evaluating four possible channel hypotheses due to ambiguity introduced by two crossover points in the CFRs, the multi-task model directly predicts the most probable hypothesis in a single forward pass. This design also enables the model to handle CFRs with more than two crossovers, which are not supported by the baseline method, making it more scalable and robust for realistic channel conditions. The multi-task model reduced inference latency by approximately 75%, confirming the computational benefit of joint inference.

While full memory profiling was not conducted, the proposed multi-task model uses a shared architecture to perform both tasks, eliminating the need for separate encoders or decoders when compared to the single-task solution. This architectural consolidation reduces the number of parameters and the memory footprint required at inference time, contributing to overall efficiency.

## 7. Conclusions

This work presents a novel approach to joint localization and channel estimation in pilot-free OFDM systems. Using uncertain channels estimated from the raw received signal, a multi-task neural network has been proposed for this task, predicting the position of the user terminal while also estimating the channel from two base stations. The solution demonstrates its efficiency across two different scenarios: a 5G simulated environment with two base stations, spread apart, and a WiFi experiment carried out in a laboratory with two antennas on the same transmitter. The study’s results show the proposed solution’s ability to outperform state-of-the-art alternatives in the studied environments for the blind channel estimation task and the localization task. By showing performance consistency, whether test measurements align with training positions or not, the method exhibits robustness and generalization ability, crucial for real-world applications.

The obtained results suggest that integrating localization with blind channel estimation within a single framework improves the localization accuracy and efficiency of the system, making the most of the spatial context provided while also reducing the number of models needed to perform both tasks. This approach improves channel estimation metrics, exploiting rich localization features for this. This combined framework opens new possibilities for efficient spectral and model resource utilization in emerging communication paradigms such as integrated sensing and communication, which demand both accuracy and adaptability in dynamic environments.

Despite the discussed advantages, the proposed solution has some limitations. One key factor limiting the performance of the solution is the channel length. While longer channels provide more expressive channel information, resulting in more accurate localization, they may not always be feasible in practical deployments, as longer channels result in a more complex channel estimation problem, highlighting the need to balance the trade-off between channel expressiveness and the constraints imposed by hardware or communication standards. Moving forward, further research could validate this approach in more diverse environments with a higher antenna density to validate the applicability of the solution. To further validate scalability, future work will investigate the model’s performance in environments with intentional interference and multiple active transmitters.

## Figures and Tables

**Figure 1 sensors-25-04095-f001:**
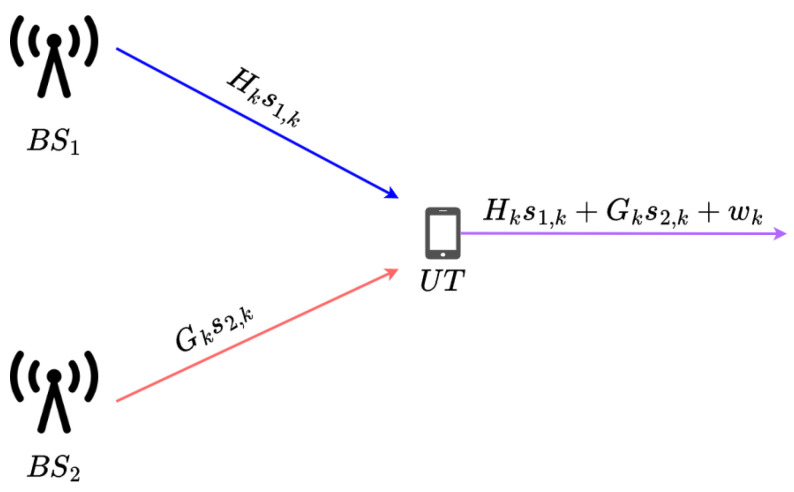
System model.

**Figure 2 sensors-25-04095-f002:**
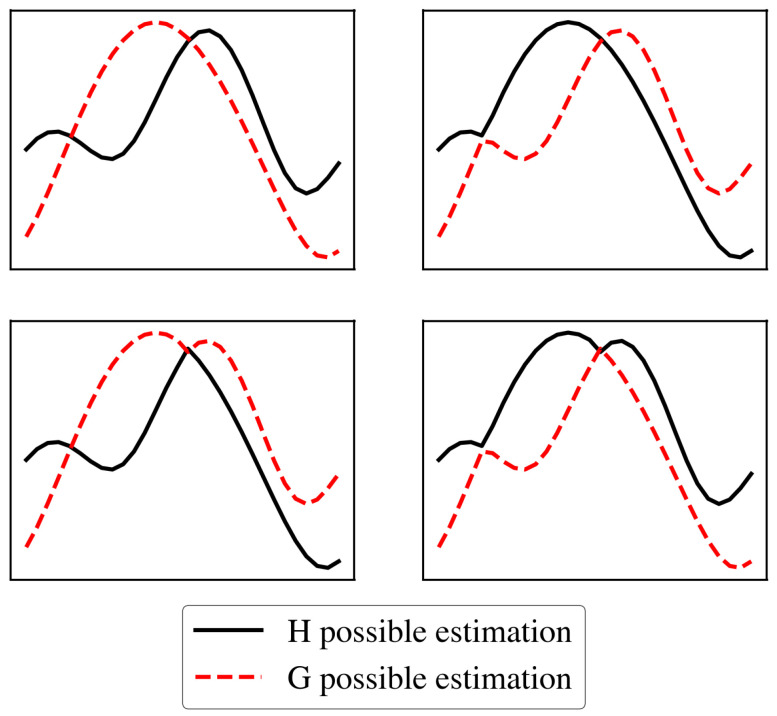
Possible combinations of H and G pair estimations.

**Figure 3 sensors-25-04095-f003:**
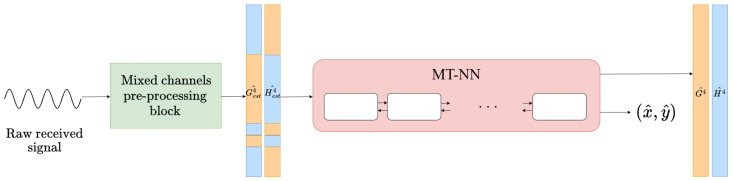
Multi-task neural network pipeline for channel estimation and localization.

**Figure 4 sensors-25-04095-f004:**
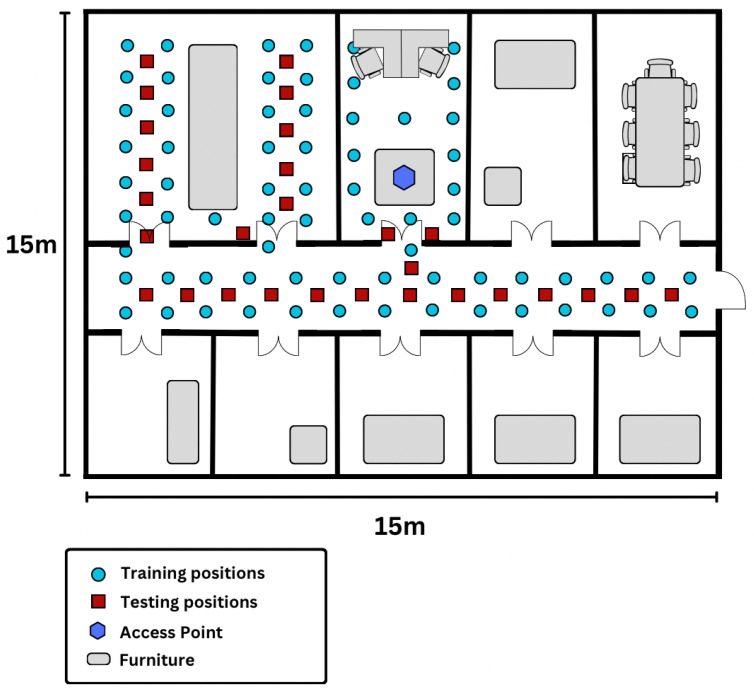
Recording positions for the WiFi localization data, as presented in [4].

**Figure 5 sensors-25-04095-f005:**
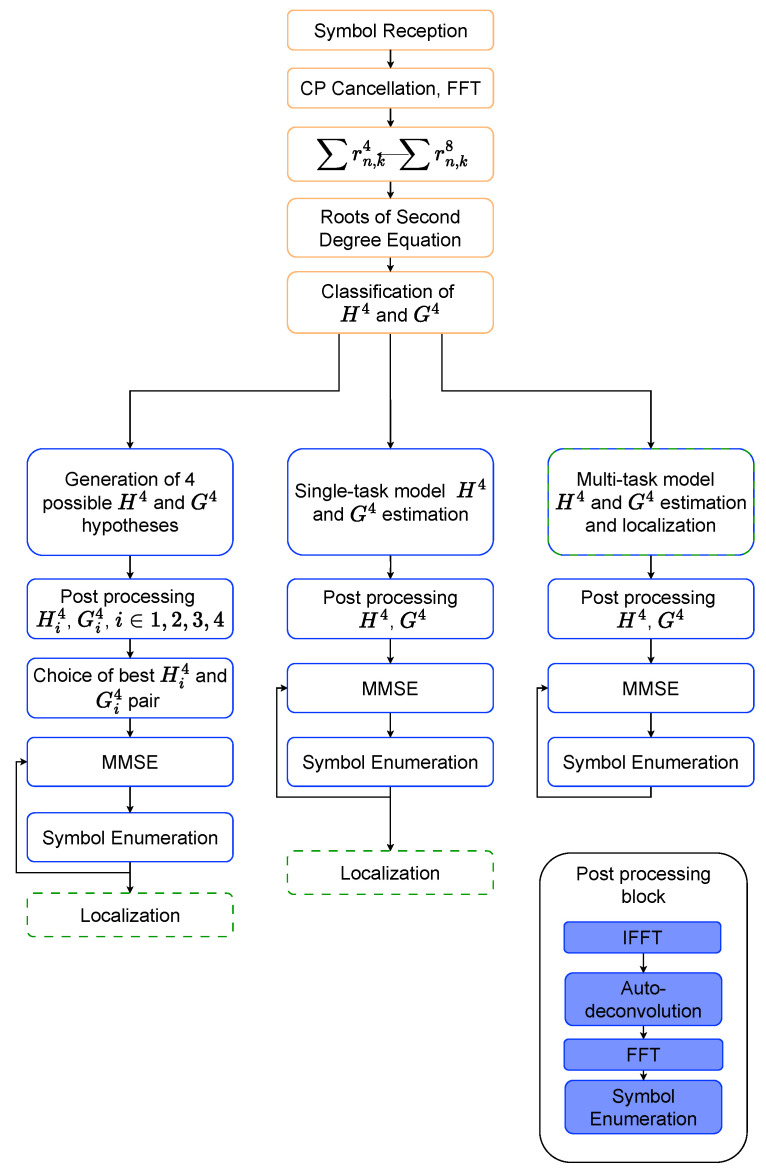
Structure of proposed solution in comparison to solution proposed in [20].

**Figure 6 sensors-25-04095-f006:**
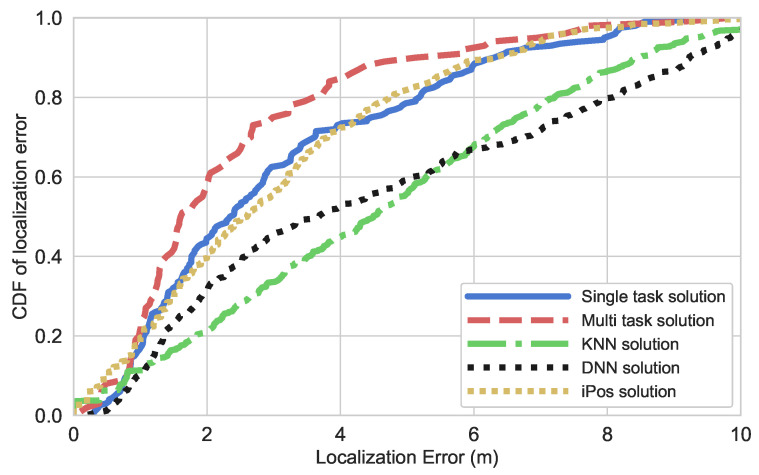
CDF of localization error for the NYUSIM scenario.

**Figure 7 sensors-25-04095-f007:**
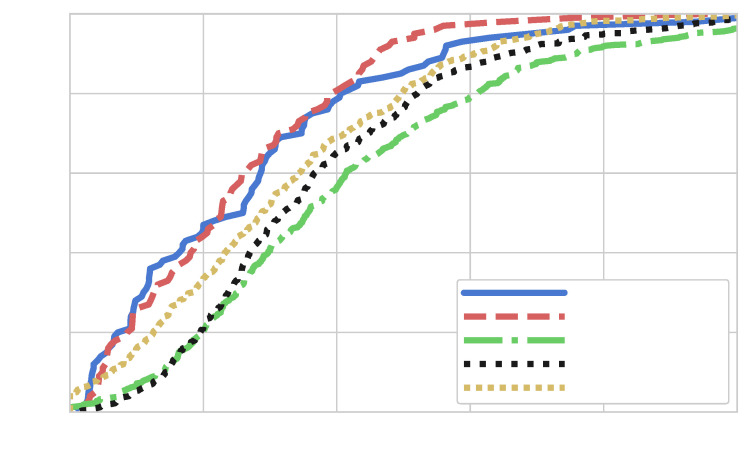
CDF of localization error for the WiFi scenario.

**Figure 8 sensors-25-04095-f008:**
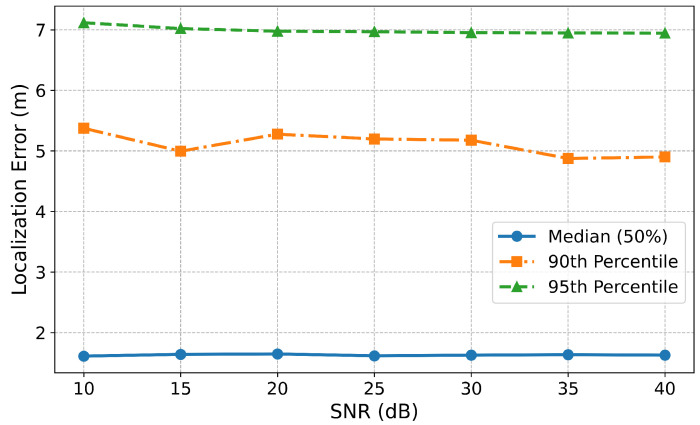
Localization error metrics for the multi-task model across SNR levels on the NYUSIM dataset.

**Figure 9 sensors-25-04095-f009:**
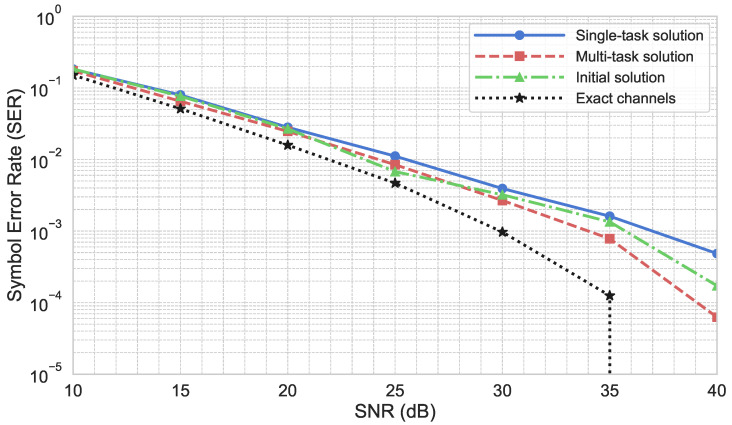
Comparison of SER results on NYUSIM dataset.

**Figure 10 sensors-25-04095-f010:**
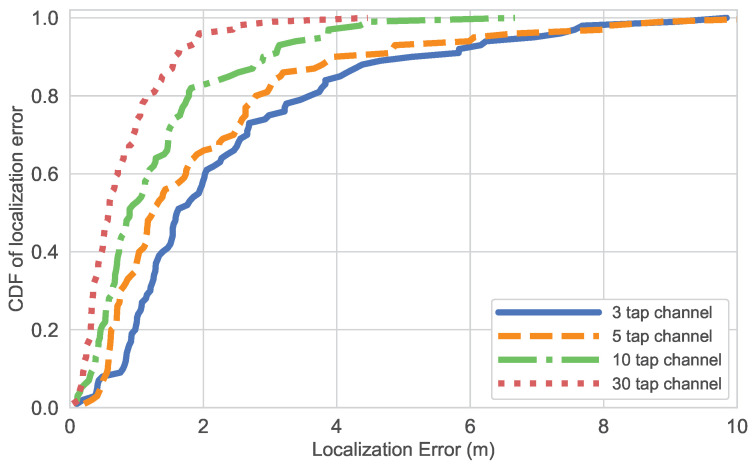
Localization error comparison for different channel lengths.

**Table 1 sensors-25-04095-t001:** NYUSIM dataset localization error metrics for single- and multi-task learning compared to SOTA at SNR = 25 dB.

Error (m)	Single-Task	Multi-Task	KNN	DNN	iPos
Min	0.33	0.09	**0.01**	0.15	**0.01**
Mean	2.76	**2.40**	4.58	3.69	3.00
Median	1.96	**1.62**	4.49	3.18	2.59
90th percentile	5.26	**5.19**	8.46	9.22	6.31

**Note:** Bold values indicate the best performance for each metric.

**Table 2 sensors-25-04095-t002:** WiFi dataset localization error metrics for single- and multi-task learning compared to SOTA at SNR = 25 dB.

Error (m)	Single-Task	Multi-Task	KNN	DNN	iPos
Min	0.11	0.15	0.01	0.15	**0.00**
Mean	2.70	**2.49**	4.09	3.69	3.16
Median	2.59	**2.27**	3.57	3.18	2.87
90th percentile	5.60	**4.61**	7.59	6.57	6.05

**Note:** Bold values indicate the best performance for each metric.

**Table 3 sensors-25-04095-t003:** NYUSIM dataset localization error metrics for different channel lengths at SNR = 25 dB.

Error (m)	3-Tap Channel	5-Tap Channel	10-Tap Channel	30-Tap Channel
Min	0.09	0.21	0.07	0.04
Mean	2.40	2.02	1.31	0.79
Median	1.62	1.24	0.89	0.58
90th percentile	5.19	4.02	2.91	1.57

**Table 4 sensors-25-04095-t004:** Inference time comparison of blind channel estimation methods (Mac M1, average per sample).

Method	Inference Time (ms/sample)
Baseline	33.90
Single-task model	8.51
Multi-task model (proposed)	**8.40**

**Note:** Bold values indicate the best performance.

## Data Availability

Dataset available on request from the authors.

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
