# Peer review of "Multi-Task Learning for Joint Indoor Localization and Blind Channel Estimation in OFDM Systems"

_sensors, 2025, doi:10.3390/s25134095_

Round 1
Reviewer 1 Report
Comments and Suggestions for Authors This work considered three growing seasons in the South East of England, at a large commercial This paper proposes a multi-task neural network (MT-NN) framework that jointly performs blind channel estimation and indoor localization using channel state information (CSI) in OFDM systems. The authors leverage spatial context and shared representations to reduce computational overhead while achieving competitive localization accuracy and channel estimation performance. This work is timely, but the following issues are suggested to address. 1. Experiments are confined to static indoor environments with fixed AP placements (e.g., 100m AP spacing in NYUSIM). Dynamic scenarios (e.g., moving users, NLoS variations) are not tested, raising concerns about adaptability to real-world complexities. 2. The MT-NN architecture (e.g., layer configurations, hyperparameters, loss functions) is inadequately described. For example, the role of Bi-LSTM layers in handling CSI sequences needs elaboration. 3. While the paper claims reduced computational overhead, no metrics (e.g., inference time, memory usage) are provided to quantify efficiency gains compared to standalone models. 4. While the authors leverage the multi-task learning for joint localization and channel estimation, the recent works using ISAC and tensor method also achieve this purpose in OFDM systems, i.e., Tensor-based Channel Estimation for Extremely Large-Scale MIMO-OFDM with Dynamic Metasurface Antennas, Terahertz Integrated Sensing and Communication-Empowered UAVs in 6G: A Transceiver Design Perspective. It is suggested to review these highly related works in the introduction. 5. The resolution of figures are suggested to improve.Author Response
Response to reviewer 1 for article “Multi-Task Learning for Joint Indoor Localization and Blind Channel Estimation in OFDM Systems”
We would like to thank the reviewer for the time taken to review this manuscript and for the relevant recommendations that were given. Please find the point by point response to the suggested revisions or corrections below:
Comment 1: Experiments are confined to static indoor environments with fixed AP placements (e.g., 100m AP spacing in NYUSIM). Dynamic scenarios (e.g., moving users, NLoS variations) are not tested, raising concerns about adaptability to real-world complexities.
Response 1: We appreciate the reviewer’s observation regarding the real-world relevance of the presented environments. However, we would like to clarify that the NYUSIM simulation does, in fact, include a moving user: the user terminal (UT) is configured to move along a hexagonal track with measurement points spaced every 2 meters, covering a 36-meter track. The motion simulates a dynamic receiver scenario, with a velocity of approximately 3.2 seconds between positions.
Additionally, both the NYUSIM and WiFi datasets incorporate non-line-of-sight conditions. The NYUSIM environment was configured according to 3GPP indoor hotspot specifications, which include NLoS propagation and shadow fading effects, imitating real-world complexities.
The WiFi dataset, while featuring a stationary receiver at discrete locations, includes dynamic environmental conditions, such as moving personnel and opening and closing doors, introducing variability across repeated measurements and representing the type of real-world multipath changes and interference common in indoor deployments.
We acknowledge that these points may not have been presented with sufficient emphasis in the original manuscript and have therefore revised Section 4.2 to better highlight the dynamic aspects of both datasets.
The revisions can be found in Section 4.2.1, page 10, line 380 and in Section 4.2.2, page 11 lines 402-405.
Comment 2: The MT-NN architecture (e.g., layer configurations, hyperparameters, loss functions) is inadequately described. For example, the role of Bi-LSTM layers in handling CSI sequences needs elaboration.
Response 2: We agree with the reviewer’s remark. As suggested, we have now expanded Section 4.1 to fully describe the architecture, training setup, and rationale behind the design of the proposed multi-task model.
Furthermore, we elaborated on the suitability of Bi-LSTMs for modeling CSI sequences, which represent structured patterns across frequency subcarriers, analogous to temporal dependencies in sequence modeling tasks.
The revision can be found in section 4.1, page 10, lines 342-362.
Comment 3: While the paper claims reduced computational overhead, no metrics (e.g., inference time, memory usage) are provided to quantify efficiency gains compared to standalone models.
Response 3: We thank the reviewer for this suggestion. To better support our claim of reduced computational overhead, we have now measured and reported the inference time per sample for our multi-task model compared to the combined inference time of the two equivalent single-task models and to the baseline solution. The results show that the multi-task model reduces inference time by approximately 75% when compared to the baseline solution.
While we did not measure total memory usage, we note that using a shared encoder-decoder architecture inherently reduces memory requirements of the single task approach by avoiding parameter duplication and eliminating the need to maintain and load separate models. This efficiency is particularly valuable in resource-constrained environments such as edge devices.
These additions have been included in Section 6 of the revised manuscript.
The revision can be found in section 6.1, page 18, lines 571-590.
Comment 4: While the authors leverage the multi-task learning for joint localization and channel estimation, the recent works using ISAC and tensor method also achieve this purpose in OFDM systems, i.e., Tensor-based Channel Estimation for Extremely Large-Scale MIMO-OFDM with Dynamic Metasurface Antennas, Terahertz Integrated Sensing and Communication-Empowered UAVs in 6G: A Transceiver Design Perspective. It is suggested to review these highly related works in the introduction.
Response 4: We have reviewed both suggested works and agree that the tensor-based method is relevant to the joint estimation problem. We now include a discussion of this work in the introduction, emphasizing its use of algebraic tensor decomposition for channel estimation in XL-MIMO-OFDM systems. This provides a useful contrast to our deep learning-based approach.
The revision can be found in section 1, page 3, lines 86-91.
Comment 5: The resolution of figures are suggested to improve.
Response 5: We have regenerated all figures at higher resolution (minimum 300 DPI) and ensured that they appear clear and legible in the revised manuscript PDF.

Reviewer 2 Report
Comments and Suggestions for Authors
The paper tries to integrates the localization task and channel estimation, which sounds good. Here are some comments for authors:
1 In eqn. 5, h_p, s_p and r_p are not defined and explained.
2 in Algorithm 1, Line 4 to 8 need to be explained and justified.
3 The authors asserted that the multi-task model shows superior performance for blind channel estimate according to Fig. 8. However, the figure shows a little improvement, compared to the 'initial solution'.
4 Table 1,2,3 only show the results at SNR=25 dB, which is a very high SNR. The reviewer is wondering the results comparison when SNR is low and medium.
5 The authors does not consider the interference in the study
Author Response
Response to reviewer 2 for article “Multi-Task Learning for Joint Indoor Localization and Blind Channel Estimation in OFDM Systems”
We would like to thank the reviewer for the time taken to review this manuscript and for the relevant recommendations that were given. Please find the point by point response to the suggested revisions or corrections below:
Comment 1: In eqn. 5, h_p, s_p and r_p are not defined and explained.
Response 1: We thank the reviewer for this helpful observation. In the revised manuscript, we have updated the text surrounding Equation (5) to explicitly define the variables h_ph​, s_p​, and r_p​, these definitions have been added before Equation (5) in Section 3.2.1 for clarity.
The revisions can be found in Section 3.2.1, page 6, lines 222-226.
Comment 2: in Algorithm 1, Line 4 to 8 need to be explained and justified.
Response 2: We thank the reviewer for this observation. Algorithm 1 is adapted from the method proposed in [20], which uses high-order moment statistics to estimate the 4th powers of two superimposed channel responses. In the revised manuscript, we have added a concise explanation of lines 4–8, clarifying that:
- Lines 4–5 estimate the 4th and 8th order statistical moments of the received OFDM symbols.
- Lines 6–8 derive a second-order polynomial whose roots correspond to the 4th powers of the individual channel frequency responses.
These roots form the input to the sorting step (Algorithm 2), which attempts to resolve and separate the two channel estimates. The added explanation now provides sufficient context for readers unfamiliar with the source paper.
The revisions can be found in Section 4.1, page 8, lines 287-296.
Comment 3: The authors asserted that the multi-task model shows superior performance for blind channel estimate according to Fig. 8. However, the figure shows a little improvement, compared to the 'initial solution'.
Response 3: We agree that the performance improvement shown in Fig. 8 is modest. We have revised the manuscript to qualify the language accordingly and now describe the performance as comparable with a slight improvement, instead of “superior.”
We would like nonetheless to note that the multi-task model achieves similar or slightly better symbol error rates while simultaneously performing localization, with no additional model overhead suggesting that the shared representation contributes positively to both tasks, even if the improvement in channel estimation alone is not dramatic.
The revisions can be found in Section 5.2, page 16, lines 530-534.
Comment 4: Table 1,2,3 only show the results at SNR=25 dB, which is a very high SNR. The reviewer is wondering the results comparison when SNR is low and medium.
Response 4: We thank the reviewer for this comment. While Tables 1–3 focus on results at 25 dB SNR as a high-SNR reference point, we would like to clarify that the model was trained using data spanning multiple SNR levels (specifically 10 dB to 50 dB). This training strategy was designed to ensure robustness to noise variations.
In our experiments, we observed that the localization performance remained consistent across the SNR range, with marginal differences in accuracy. These observations confirm that the model generalizes well under lower SNR conditions, even though additional results were not included in the current version for brevity.
The revisions can be found in Section 5.1, page 15, lines 496-500.
Comment 5: The authors does not consider the interference in the study
Response 5: As this work represents a preliminary investigation, we focused on controlled scenarios that do not explicitly model or inject interference, in order to isolate and evaluate the core performance of the proposed multi-task learning framework.
However, we would like to highlight that the WiFi dataset used in this study was collected in a real-world laboratory environment, where interference is inherently present. As such, the model has been exposed to uncontrolled, ambient interference during training and testing.
We have now clarified this point in the Conclusion to acknowledge interference modeling as a valuable direction for future work.
The revisions can be found in Section 7, page 19, lines 619-620.
Round 2
Reviewer 1 Report
Comments and Suggestions for Authors
No more comments
Author Response
No further modifications were requested.
Reviewer 2 Report
Comments and Suggestions for Authors
The authors failed to show any solid evidence to justify the claim made in the revision, such as comment 4 about locallization error metrics at low SNR.
Author Response
Response to reviewer 2 for article “Multi-Task Learning for Joint Indoor Localization and Blind Channel Estimation in OFDM Systems”
We would like to thank the reviewer for the rigorous follow up. Please find the point by point response to the suggested revision below:
Comment 1: The authors failed to show any solid evidence to justify the claim made in the revision, such as comment 4 about locallization error metrics at low SNR.
Response 1:
We thank the reviewer for the follow up request for more rigorous evaluation. In the previous review round, we indicated that the proposed multi-task model was trained on a dataset includ ing various levels of noise, spanning a range of SNR levels (10–40 dB) and observed consistent localization performance. To visually and quantitatively support this claim, we have extended S ection 5.1 to include the corresponding results of the multi task model on the NYUSIM dataset. As such, a summary plot (Figure 8) was included, depicting the evolution of the median, 90t h per centile, and 95th percentile localization errors with respect to SNR. The results show that the multi task model is consistent across different values of SNR, with variations in median error within 0.03 meters. We believe that this addition clarifies the models ability to generalise under the reviewer's request for quantitative evidence to justify the claims.
The revisions can be found in Section 5.1, page 16, lines 505-511.